# OpenReview forum: "The Devil is in the Neurons: Interpreting and Mitigating Social Biases in Language Models"
_ICLR.cc/2024/Conference — ICLR 2024 poster_

### Official Review · Reviewer_auwC · 2023-10-20

**Soundness:** 4 excellent
**Presentation:** 4 excellent
**Contribution:** 4 excellent
**Rating:** 8
**Confidence:** 3

**Summary:**

The paper presents an interesting approach called Integrated Gap Gradients that aims to identify the neurons in a large language model that are responsible for the exhibition of undesirable behavior like social bias. Also, the paper presents a technique called Bias Neuron Suppression that aims to suppress the detected neurons that are responsible for the undesirable behavior, hence mitigating social biases in large language models. The paper performs an extensive evaluation comparing the approach to multiple baselines, demonstrating the merits of the proposed approach in mitigating social biases and being more fair.

**Strengths:**

The paper focuses on an important issue, and I believe that the proposed approaches can have a significant impact on the research community that aims to study social biases and ways to mitigate these issues in large language models. The proposed approach is a significant step toward more interpretable Artificial Intelligence and aims to mitigate social biases without the need for expensive re-training or fine-tuning of very large language models like BERT or RoBERTa. The paper makes an extensive evaluation of the proposed approach and compares it with multiple baselines such as DPCE, AutoDebias, and Union_IG, demonstrating that the proposed approach outperforms them in terms of various bias metrics. Also, I really liked that the paper makes an extensive evaluation, including a large number of demographic attributes as well as modifiers. Overall, I believe that this is a good paper submission, and I believe it can advance research and state-of-the-art methods in mitigating social biases that arise from the use of large language models.

**Weaknesses:**

Overall, I do not have major concerns with the paper, and I have only some mainly minor issues/suggestions for the authors to further improve their paper. First, it will be good to clarify how you came up with the Demographic dimensions that are included in Table 1. Is this based on previous work or did you construct these dimensions for the purposes of this work? Also, for figure 2, the paper does not clarify what model is used to generate the results? Is this based on the BERT model? I suggest to the authors to clarify this and also discuss if they observe any differences in the presented results for Figure 2 across models. Finally, I would have liked to see a more detailed discussion and explanation of how the authors envision the use of the proposed approaches in other applications beyond social biases and debasing. The authors touch briefly upon this in the Introduction, however, they do not go into details, and it is unclear from a reader’s perspective how the proposed approaches can be applied for other purposes.

**Questions:**

1. How are the demographic dimensions in Table 1 constructed?
2. How do you envision the proposed approaches to be applied for other purposes beyond debiasing?

---

> ### Author Response · Authors · 2023-11-23
> **Rebuttal by Authors**
>
> We would like to thank the reviewer for the constructive comments! The following are our detailed responses regarding all major concerns. We hope the following responses can clarify the missing points and address these concerns.
>
> **Q1: It will be good to clarify how you came up with the Demographic dimensions that are included in Table 1. Is this based on previous work or did you construct these dimensions for the purposes of this work?**
>
> **A1**:
> We constructed the demographic dimension by concluding from previous works [1,2] in social bias research.
> Most previous research concentrated on the few prevalent demographic dimensions, such as gender or ethnicity. Besides, previous research mainly conducted their research on only one or two demographic dimensions. To be more comprehensive, we collect 9 demographic dimensions of concern in the current fairness research field from previous works.
>
> &nbsp;
>
> [1] Gender and Representation Bias in GPT-3 Generated Stories.
>
> [2] From Pretraining Data to Language Models to Downstream Tasks: Tracking the Trails of Political Biases Leading to Unfair NLP Models.
>
> **Q2: The paper does not clarify what model is used to generate the results for figure 2.**
>
> **A2**:
> Thank you for your suggestion! The results are generated by FairBERTa. We have also reached similar conclusions for other models (BERT and RoBERTa). We have clarified this in our revised paper.
>
> **Q3: Would have liked to see a more detailed discussion and explanation of how the authors envision the use of the proposed approaches in other applications beyond social biases and debasing.**
>
> **A3**
> Thank you so much for pointing this out! Our IG$^2$ can be applied to other research fields, e.g., comparative knowledge study. Previous works have found that models trained to learn comparative knowledge have better reasoning ability compared with those trained on singular knowledge. For example, models trained to learn that ''moon is round, not square'' can perform better in reasoning tasks than models trained to learn that ''moon is round''. Our IG$^2$ can be applied to detect neurons contributing to storing comparative knowledge and then amplify the activation of these neurons, in order to achieve better reasoning ability.

---

### Official Review · Reviewer_pmu7 · 2023-10-29

**Soundness:** 3 good
**Presentation:** 4 excellent
**Contribution:** 4 excellent
**Rating:** 8
**Confidence:** 4

**Summary:**

The paper introduces a new method, Integrated Gap Gradients (IG2), for measuring how much an individual neuron contributes to differences between logit predictions at the outputs. Combined with curated prompts that have positive or negative sentiment and are intended to be completed with a demographic group, this method is used to identify neurons in the network that most strongly contribute to differences between demographic groups along different dimensions, including gender, ethnicity, etc. Once these neurons are identified, the paper applies a new technique, Bias Neuron Suppression (BNS), to zero out the activations of neurons that are above a threshold score. The paper shows that when identified bias neurons are amplified, demographic gaps increase, and when they are suppressed, demographic gaps decrease. The paper also shows improvement on scores based on StereoSet for measuring the demographic bias of a language model. Finally, the paper also presents interesting empirical results comparing to other debiasing methods, showing for example that the number of bias neurons in FairBERTa does not decrease. Instead, the neurons get shifted into earlier layers compared to networks that have not been debiased. Overall, the paper demonstrates the promise of this new technique.

**Strengths:**

The paper’s primary strengths are the empirical results and the clarity of the new method proposed. The work is an interesting extension of the integrated gradients technique to measurements of demographic bias, and the results seem to support the efficacy of the method well. The work is clearly written and easy to follow as well.

**Weaknesses:**

The primary weakness of the work is the fact that the paper primarily relies on StereoSet and associated metrics to measure the effectiveness of the debiasing technique. While this is not a major limitation, it might have been nice to evaluate on another dataset with targeted queries for a specific group, such as WinoQueer (https://arxiv.org/abs/2306.15087). Another weakness of the work is a more thorough discussion of the limitations of the method, and I have some questions in the following section that could be used to expand this discussion.

**Questions:**

Figure 2: it’s unclear to me from the caption what exactly the x-axis is plotting. Can the authors clarify?

Table 6: Could use a bit more explanation about what the “inter” and “intra” columns represent

Questions about limitations

Is there any empirical evidence or motivation, aside from the results here, to support the assumption that single neurons can strongly contribute to demographic disparities?

Is there a way to generalize the method to take into account interactions or correlations between neurons that might affect the result?
Do you think the method is limited by the kinds of prompts that you have templated here? How would you generalize?

Can this method be extended beyond differences in sentiment for demographic groups?

---

> ### Author Response · Authors · 2023-11-23
> **Rebuttal by Authors (1/2)**
>
> We would like to thank the reviewer for the constructive comments! The following are our detailed responses regarding all major concerns. We hope the following responses can clarify the missing points and address these concerns.
>
> **Q1: The primary weakness of the work is the fact that the paper primarily relies on StereoSet and associated metrics to measure the effectiveness of the debiasing technique. It might have been nice to evaluate on another dataset, such as WinoQueer.**
>
> **A1**:
> Thank you for your advice! We have conducted the experiments following your advice on WinoQueer.
>
> | Model                | WinoQueer Overall | LGBTQ | Queer | Transgender | NB    | Bisexual | Pansexual | Lesbian | Asexual | Gay   |
> |----------------------|-------------------|-------|-------|-------------|-------|----------|-----------|---------|---------|-------|
> | BERT-Base-cased      | 64.40             | 91.55 | 58.53 | 91.72       | 78.93 | 43.01    | 27.33     | 90.97   | 33.44   | 41.71 |
> | BERT-Base-cased + BNS (Ours) | 60.38   | 88.04 | 53.85 | 89.16       | 75.42 | 39.36    | 24.05     | 86.51   | 31.92   | 46.58 |
> | RoBERTa-Base         | 69.18             | 74.17 | 61.68 | 49.04       | 87.93 | 67.1     | 85.91     | 81.27   | 81.63   | 62.19 |
> | RoBERTa-Base + BNS (Ours) | 66.74     | 72.81 | 60.22 | 47.79       | 86.04 | 64.80    | 84.32     | 77.69   | 80.17   | 60.01 |
> | FairBERTa            | 68.87             | 73.92 | 61.54 | 48.99       | 87.45 | 66.92    | 85.88     | 81.13   | 81.05   | 61.68 |
> | FairBERTa + BNS (Ours) | 66.51          | 71.48 | 60.35 | 47.62       | 86.25 | 64.27    | 84.19     | 80.07   | 79.68   | 59.93 |
>
> **Q2: More clarification on the x-axis in Figure 2.**
>
> **A2**:
> The x-axis is the randomly selected Judged Unfair Targets (JUTs). The definition of the JUTs is presented in Section 2.3. "-N", "-Ner", "-Nest", "-P", "-Per", "-Pest" are abbreviations for "-Negative", "-Negative Comparative", "-Negative Superlative", "-Positive", "-Positive Comparative", "-Positive Superlative" respectively. We have added more explanations in the caption for clearer expression. We have also added the abbreviations for different types of modifiers in Table 2 to make our paper more readable.
>
> **Q3: More explanation about what the “inter” and “intra” columns represent in Table 6.**
>
> **A3**:
> (1) **The ''intra'' column**. We counted the number of detected social bias neurons shared by different prompt templates within the same JUT. For example, we counted the number of detected social bias neurons shared among prompt templates within the JUT of (lonely, Black, White). We computed this metric for all the JUTs for a certain demographic dimension (e.g., "Ethnicity") and then reported the average. The relatively large values in this column reveal that our IG$^2$ is effective and robust in detecting the social bias neurons without interference from different grammar and sentence structures.
>
> (2) **The "inter" column**. We counted the number of detected social bias neurons shared across different JUTs. For example, we counted the number of detected social bias neurons shared between JUT of (lonely, Black, White) and JUT of (lonely, Hispanic, American) using prompt template #1 (prompt templates are defined in Table 3). We randomly selected some prompt templates and paired JUTs to gather this data. The relatively small values in this column reveal that different types of social biases share almost no social bias neurons. This suggests that our IG$^2$ identifies exclusive social bias neurons for different types of social biases.

---

> ### Author Response · Authors · 2023-11-23
> **Rebuttal by Authors (2/2)**
>
> **Q4: Is there any empirical evidence or motivation, aside from the results here, to support the assumption that single neurons can strongly contribute to demographic disparities?**
>
> **A4**:
> Our motivation comes from the field of neuroscience and its understanding of the working principles of the human brain. The human brain primarily relies on neurons to store information and uses the electrical signals within neurons for information transmission and processing (the process of thinking). Considering the origin of neural networks in computer science arises from the study of neuroscience, we speculate whether this might be similar for neural networks. We wonder if those biased cognitions against marginalized groups might also be stored in some neurons. Some literature has also conducted studies on the behavior of neurons. [1] finds that some classes were represented more selectively than others within networks, suggesting that certain neurons might have a stronger association with specific classes. [2] shows that certain neurons may become specialized in detecting specific features or concepts, where the specialization can potentially extend to features related to social biases. These findings underscore the potential impact of individual neurons on the biases and overall behavior of neural networks, offering empirical backing for the research of locating specific neurons to address biases in language models.
>
> &nbsp;
>
> [1] Ablation Studies in Artificial Neural Networks.
>
> [2] A Survey on Neural Network Interpretability.
>
> **Q5: Is there a way to generalize the method to take into account interactions or correlations between neurons that might affect the result?**
>
> **A5**:
> Thank you for your very insightful question! This is a very good and valuable question!
> First, quick answer: there should be an appropriate way to generalize both neurons and the interactions among neurons. In fact, we have considered dealing with the interactions among neurons and are currently working on a new project of studying and manipulating these interactions. In contrast to neurons, we refer to these interactions as ''edges''. There are many patterns of edges connecting neurons, e.g., string patterns, square patterns, etc. We believe that by precisely controlling the edges that connect neurons, we can achieve more ideal debiasing results: more accurately blocking the storage and propagation of biases while minimally impacting the model's capabilities. Furthermore, we have discovered that there is a mathematical form that can abstract and generalize neurons and edges: generalizing them into a homogenized minimal unit of neural networks.
>
> However, we hope you can understand that we have not mentioned these contents for the sake of clarity in our writing. There are many types of connection patterns for edges, and controlling these edges also involves the study of the propagation and flow of biases among neurons. This part also requires a significant amount of visualization, within which many interesting findings emerge. However, it is impossible to discuss all these contents in one work comprehensively. Therefore, to avoid ambiguity, we have not mentioned the interaction relationships between neurons in this work. To be honest, we are very surprised and excited to see this question. It also gives us much confidence to continue exploring the interactions among neurons. We will further explore this interesting question in the new project and hope to design an appropriate methodology to generalize neurons and edges (interactions among neurons). We hope our work can bring further inspiration to fairness research. Thank you again for your insightful question!
>
> **Q6: Is the method limited by the kinds of prompts in the paper? How would you generalize?**
>
> **A6**:
> We could expand our approach to include a broader variety of prompt structures that encapsulate different linguistic styles and contexts. Here's an example of a new template: "In a situation where a [Demographic Dimension] individual is [Context], they are often perceived as [MASK]." The [Context] slot can be filled with various scenarios or actions, for example, "leading a team," "making a decision," or "in a family setting." This would help in capturing a wider spectrum of potential biases.
>
> **Q7: Can this method be extended beyond differences in sentiment for demographic groups?**
>
> **A7**:
> Our IG$^2$ can be applied to other research fields, e.g., comparative knowledge study. Previous works have found that models trained to learn comparative knowledge have better reasoning ability compared with those trained on singular knowledge. For example, models trained to learn that ''moon is round, not square'' can perform better in reasoning tasks than models trained to learn that ''moon is round''. Our IG$^2$ can be applied to detect neurons contributing to storing comparative knowledge and then amplify the activation of these neurons, in order to achieve better reasoning ability.

---

### Official Review · Reviewer_zYP2 · 2023-10-30

**Soundness:** 3 good
**Presentation:** 3 good
**Contribution:** 3 good
**Rating:** 6
**Confidence:** 3

**Summary:**

This paper investigates the suitability of interpretability techniques used to attribute individual neurons to model generations for the study of social bias in neural networks. To do so, the authors introduce Integrated Gap Gradients (IG2) as an approach to identify neurons attributable to social bias. Their approach is evaluated on a dataset of around 200000 template-generated samples. The evaluation is carried out by either suppressing or amplifying the detected neurons, and the experimental results demonstrate that doing so indeed reduces or increases the generation of biased outputs. Based on these results, the authors propose a debiasing approach based on the suppression of detected neurons and evaluate it on the StereoSet benchmark. Comparing this approach to a range of baselines, the authors show that the method successfully debiases model predictions while at the same time having little impact on unrelated model predictions (measured via the Language Modeling Score).

**Strengths:**

* The paper combines interpretability techniques with an application to social bias research in neural networks and shows that doing so yields valuable insights.
* The results are promising and might represent a foundation for future work applying the method to other applications in NLP.
* The paper is well-written and the experiments are overall thorough.

**Weaknesses:**

* The paper could spend more attention on the implications of the presented results. Since the authors mention the importance of fairness research in this context, a brief discussion on the usability, practicability, and limitations of their results would be desirable.
* The paper does not discuss potential implications of BNS on other tasks that the investigated LMs are faced with. While this is difficult to measure directly, it remains unclear to what extent the manipulation of neurons at inference time impacts model performance on other tasks. I’d recommend the authors to address this in the manuscript carefully.

**Questions:**

* Did you experiment with regularizing detected neurons rather than suppressing them entirely? Considering their suppression more gradually as opposed to zero-ing them out might help in increasing the Language Modeling Score while keeping a high Stereotype Score.
* Not really a question but more a suggestion: I’d recommend moving the introduction of BNS to after the first report of experiments to improve the flow of the paper.
* Would it be worth adding a few examples (rather than only templates) to Table 3?

---

> ### Author Response · Authors · 2023-11-23
> **Rebuttal by Authors**
>
> We would like to thank the reviewer for the constructive comments! The following are our detailed responses regarding all major concerns. We hope the following responses can clarify the missing points and address these concerns.
>
> **Q1: A brief discussion on the usability, practicability, and limitations of their results would be desirable.**
>
> **A1**:
> Thank you for your advice! We briefly discuss the implications, practicability and limitations in the following.
> - Implications of the presented results. We propose an interpretable method to analyze social bias in PLMs. Through our experiments, we found that previous debiasing methods, represented by FairBERTa, do not significantly reduce the number of bias neurons but merely alter their distribution (as shown in Figure 3). We believe this finding could bring new insights to the field and inspire other researchers to design more effective debiasing methods.
> - Usability/Practicability. We foresee our findings impacting the development of more equitable language models. On one hand, our method offers good interpretability, providing insights into how to design more efficient debiasing methods. On the other hand, our method can be combined with previous debiasing approaches (such as FairBERTa) to further enhance effectiveness. Mitigating social biases in PLMs also has extensive practical applications. For example, reducing biases in resume screening and job matching, and promoting fair hiring practices.
> - Limitations. While our study provides valuable insights, we recognize there exist limitations. For example, our proposed BNS method directly sets the activation values of selected social bias neurons to zero. Although this is effective, designing a more refined suppression method might yield even better results. These present opportunities for future research.
>
> We will add a section discussing the implications, practicability, and limitations of our findings in the revised paper.
>
> **Q2: The paper does not discuss potential implications of BNS on other tasks that the investigated LMs are faced with.**
>
> **A2**:
> Thank you for the insightful comment! We follow your suggestion and further conduct experiments on the SST-2 dataset (one of GLUE tasks). We report results on the development set of SST-2. The result of RoBERTa-Base is taken from the corresponding paper.
> |Method| SST-2 |
> |-----------------------|-------------|
> |RoBERTa-Base | 92.90 |
> |RoBERTa-Base + BNS (ours) | 92.38 |
>
> **Q3: Did you experiment with regularizing detected neurons rather than suppressing them entirely?**
>
> **A3**:
> We are very grateful for your suggestion, which is very enlightening for our future work! During the rebuttal, we follow your suggestion and conduct a quick experiment on BERT-Base-cased. We adjust the activation values of the detected bias neurons to 0.5 times their original values, instead of directly setting them to zero. We report the results in the following table.
>
> | Suppression Ratio | SS->50.00 | LMS (↑) | ICAT (↑) |
> |-------------------|-----------|---------|----------|
> | 0                 | 52.78     | 86.64   | 81.82    |
> | 0.5               | 52.95     | 87.08   | 81.94    |
>
>
> **Q4: Recommend moving the introduction of BNS to after the first report of experiments to improve the flow of the paper.**
>
> **A4**:
> Thank you for your advice! We have modified our paper accordingly.
>
> **Q5: Would it be worth adding a few examples (rather than only templates) to Table 3?**
>
> **A5**:
> Thank you for your advice! We have followed your advice and added the example in the revised paper.

---

> > ### Comment · Reviewer_zYP2 · 2023-11-23
> > **Thanks!**
> >
> > Thank you for addressing my comments and providing those additional results. It looks like the results especially from Q3 could be analyzed further, and I encourage the authors to add such experiments into the paper. I will keep my rating fixed and am in favor of acceptance.

---

### Official Review · Reviewer_GcZ6 · 2023-10-30

**Soundness:** 3 good
**Presentation:** 3 good
**Contribution:** 3 good
**Rating:** 6
**Confidence:** 4

**Summary:**

The paper proposes INTEGRATED GAP GRADIENTS (IG2) approach that identifies problematic neurons in a language model that contribute to undesirable behavior, such as bias. They then use this identification framework and propose BIAS NEURON SUPPRESSION (BNS) to mitigate the effect of these problematic neurons to reduce the observed biases in LLMs. Authors also perform experiments to showcase effectiveness of their detection and mitigation approaches.

**Strengths:**

1. The paper is easy to follow and is written clearly.
2. The approach is easy to implement and be applied.
3. The paper studies an important issue.

**Weaknesses:**

1. The technical contribution of the paper is not significant as it simply extends a previous existing approach for identification of problematic neurons in the context of bias and fairness.
2. The experiments are done only on masked language modeling tasks which can limit the study.
3. The approach operates between binary demographics and might be limited when we want to study biases amongst large pool of fine-grained demographics.
4. Their constructed dataset only covers few demographics.
5. The paraphrases for constructing the dataset are generated using a language model which can impose its own biases in the study.

**Questions:**

For results in Table 5 under FairBERTa why are some approaches missing? Some clarification on this can be helpful.

---

> ### Author Response · Authors · 2023-11-23
> **Rebuttal by Authors**
>
> We would like to thank the reviewer for the constructive comments! The following are our detailed responses regarding all major concerns. We hope the following responses can clarify the missing points and address these concerns.
>
> **Q1: The technical contribution of the paper is not significant.**
>
> **A1**:
> We are not "simply extending a previous existing approach in the context of bias and fairness," but have a very clear motivation. In the field of social bias, previous methods mainly require finetuning or even pre-training of PLMs, which is highly costly. We aim to address this issue by proposing an interpretable and efficient method to locate and reduce social biases. Drawing inspiration from the IG approach, we designed our IG$^2$. Our IG$^2$ differs from the IG method: IG is used to analyze singular knowledge attribution, but our method analyzes multiple knowledge. Other reviewers also acknowledge our contribution to the community, for instance, “*The results are promising and might represent a foundation for future work applying the method to other applications in NLP*.” (Reviewer zYP2), and “*I believe that the proposed approaches can have a significant impact on the research community that aims to study social biases and ways to mitigate these issues in large language models*.” (Reviewer auwC).
>
> **Q2: The experiments are done only on masked language modeling tasks which can limit the study.**
>
> **A2**:
> Our method can be applied to other tasks, e.g., autoregressive generation. For example, we could use a prompt such as "*Please complete the following sentence. The gender of this handsome person is*". Then we could analyse the logit gap of the first token predicted by the language model. We will further explore this extension in the near future.
>
> **Q3: The approach operates between binary demographics and might be limited when study biases amongst large pool of fine-grained demographics.**
>
> **A3**:
> We can extend our method to a large pool of fine-grained demographics. The logit gap between pairwise demographics is the most direct metric to reveal the unfairness for different demographics. In fact, to support the attribution of the distribution gap for more than two demographics, we can also choose other metrics that can be applied to measure the degree of dispersion among multiple demographics, such as standard deviation.
>
> **Q4: Their constructed dataset only covers few demographics.**
>
> **A4**:
> To study social bias more systematically, we constructed our dataset with 19 pairs of demographics, totaling 38 demographics, even including socioeconomic status, which is rarely studied in existing fairness studies. We believe the demographics used in the paper cover a wide enough range (this is recognized by Reviewer auwC: "*I really liked that the paper makes an extensive evaluation, including a large number of demographic attributes as well as modifiers*"). We want to point out that previous works [1,2,3,4,5,6] often only analyze one or two demographic dimensions.  However, we understand the concerns of the reviewer and will consider adding more demographics in the future to make the work more comprehensive.
>
>  &nbsp;
>
> [1] Mitigating Political Bias in Language Models Through Reinforced Calibration.
>
> [2] Investigating Gender Bias in Language Models Using Causal Mediation Analysis.
>
> [3] The Woman Worked as a Babysitter: On Biases in Language Generation.
>
> [4] From Pretraining Data to Language Models to Downstream Tasks: Tracking the Trails of Political Biases Leading to Unfair NLP Models.
>
> [5] Gender and Representation Bias in GPT-3 Generated Stories.
>
> [6] Language Models Get a Gender Makeover: Mitigating Gender Bias with Few-Shot Data Interventions.
>
> **Q5: The paraphrases for constructing the dataset are generated using a language model which can impose its own biases in the study.**
>
> **A5**:
> We use a language model to generate templates with placeholders (such as ([Demographic_Dimension], [Modifier], and [MASK])) only, without including any specific demographics, as shown in Table 3. Therefore, this process does not introduce any biases from the language model itself.
>
> **Q6: Why are some approaches missing for FairBERTa in Table 5?**
>
> **A6**:
> We guess that you are referring to the absence of "FairBERTa+DPCE" and "FairBERTa+AutoDebias" in the table. FairBERTa, unlike the other two baseline models (BERT-Base-cased and RoBERTa-Base), is a debiasing method based on RoBERTa-Base. Therefore, we did not combine it with DPCE or AutoDebias. We combined BNS with FairBERTa in the table to demonstrate that our method is not only superior to FairBERTa alone but also achieves better results when combined, proving the plug-and-play capability of our approach.

---

### Official Review · Reviewer_jE7C · 2023-11-01

**Soundness:** 3 good
**Presentation:** 3 good
**Contribution:** 3 good
**Rating:** 6
**Confidence:** 4

**Summary:**

This paper aims to understand why the bias happens in LMs. They use integrated gap gradients (IG^2) to identify which neurons in language models contribute to the social bias. With the identified neurons, the authors proposed bias neuron suppression method to reduce the bias in LMs. Experimenting with BERT, RoBERTa and FairBERTa, the proposed IG^2 method demonstrates to reduce the bias.

**Strengths:**

Extending IG method to understand the biases in language models, where the topic is interesting and important. The authors did a good job of structuring the paper, so it is easy to follow.

**Weaknesses:**

It seems the IG^2 requires calculating the gap between the logits of different attributes. I'm wondering how general will it be, e.g., how can we extend it to other tasks instead of predicting the sensitive attributes? For example, what if the task is a hate speech detection?

**Questions:**

- Does the method only work when the output of the model is to predict the sensitive attributes (so that you can calculate the logit gap)?
- What model is being examined in Fig.2? Does the conclusion hold for all models, especially for FairBERTa?
- It seems most of the time, Union_IG can make the bias score lower. Is it because Union_IG contains more neurons? Does this mean adding more neurons generally helps reduce the bias? How sensitive is the result w.r.t. $\sigma$?
- What if you replace Union_IG to Intersection_IG, i.e., the intersection of bias neurons for different attributes?
- Table 5, it should be "StereoSet".

---

> ### Author Response · Authors · 2023-11-23
> **Rebuttal by Authors (1/2)**
>
> We would like to thank the reviewer for the constructive comments! The following are our detailed responses regarding all major concerns. We hope the following responses can clarify the missing points and address these concerns.
>
> **Q1: Does the method only work when the output of the model is to predict the sensitive attributes (so that you can calculate the logit gap)? How can we extend it to other tasks instead of predicting the sensitive attributes?**
>
> **A1**: The proposed methods can be extended to other fields and tasks (e.g., comparative knowledge study) without predicting sensitive attributes. Previous works have found that models trained to learn comparative knowledge have better reasoning ability compared with those trained on singular knowledge. For example, models trained to learn that ''moon is round, not square'' can perform better in reasoning tasks than models trained to learn that ''moon is round''. Our $IG^2$ can be applied to detect neurons contributing to storing comparative knowledge and then amplify the activation of these neurons, in order to achieve better reasoning ability.
>
> **Q2: How can we extend it to hate speech detection?**
>
> **A2**:
> We think the proposed method could not be directly applied to the hate speech detection task. As far as we know, the target of the hate speech detection task is to design a detection model and judge whether some given sentences contain toxic or discriminative expressions. Biases or toxicity are not generated by the detection model, but exist in the text input. In contrast, our method is to attribute biased output to model internal neurons. This requires that biased sentences are generated by the model we focus on.
>
> **Q3: What model is being examined in Fig.2? Does the conclusion hold for all models, especially for FairBERTa?**
>
> **A3**:
> The model examined in Fig.2 is FairBERTa. We have also reached similar conclusions for other models (BERT and RoBERTa). We have revised the caption to make it clearer.
>
> **Q4: It seems Union_IG can make the bias score lower. Is it because Union_IG contains more neurons? Does this mean adding more neurons helps reduce the bias? How sensitive is the result w.r.t. $\sigma$?**
>
> **A4**:
> (1) No. Sometimes Union_IG contains fewer neurons than $IG^2$. The reason for Union_IG making the bias score lower is that Union_IG contains all the neurons that affect the learned knowledge about different demographics. Therefore, when we suppress all the neurons detected by Union_IG, the model forgets almost all the knowledge about these demographics. In other words, the model is hard to identify these demographics in this circumstance. For example, the model may not know what is "Black" or "White". This will severely impair the language model’s capability.
>
> (2) Yes, adding more bias neurons helps reduce social biases, but at the cost of sacrificing the capabilities of language models.
> Therefore, we need to strike a balance between reducing social bias and maintaining the capabilities of language models.
>
> (3) The result is not very sensitive to $\sigma$. We report the ablation studies on BERT in the following table.
>
> | \sigma       | SS->50.00 | LMS (↑) | ICAT (↑) |
> |--------------|-----------|---------|----------|
> | 0.1*maximum  | 52.13     | 85.01   | 81.39    |
> | 0.2*maximum  | 52.78     | 86.64   | 81.82    |
> | 0.3*maximum  | 52.99     | 86.77   | 81.58    |
> | 0.4*maximum  | 53.15     | 86.82   | 81.35    |

---

> ### Author Response · Authors · 2023-11-23
> **Rebuttal by Authors (2/2)**
>
> **Q5: Replace Union_IG to Intersection_IG.**
>
> **A5**:
> We follow your suggestion and conduct additional experiments with Intersection_IG. We first study the impact of amplifying or suppressing the neurons detected by Intersection_IG on the logits gap (an experiment similar to that in Figure 2). From the table below, we find that neurons detected by Intersection_IG do not have direct relations with the manifestation of social biases. For example,  both amplifying and suppressing neurons lead to a decreased ratio in the logits gap of "Physical Appearance-P".
>
> | JUT                   | Suppression | Amplification |
> |-----------------------|-------------|---------------|
> | Gender-N              | 19.62       | -6.07         |
> | Gender-Ner            | 4.13        | -0.26         |
> | Gender-Nest           | -1.35       | -3.82         |
> | Gender-P              | 0.06        | 7.49          |
> | Gender-Per            | -10.45      | -16.10        |
> | Gender-Pest           | -0.96       | -0.08         |
> | Physical Appearance-N | -5.99       | 0.12          |
> | Physical Appearance-Ner | 8.97      | 10.41         |
> | Physical Appearance-Nest | -0.93    | -0.48         |
> | Physical Appearance-P | -13.74      | -20.19        |
> | Physical Appearance-Per | 15.52     | 8.83          |
> | Physical Appearance-Pest | -6.08    | -0.49         |
>
> We also present the results on Stereoset as below. We can see that after suppressing neurons detected by Intersection_IG, the SS score gets even worse. This indicates that social biases have not only failed to be mitigated, but it has also become more severe.
>
> | Model                         | SS->50.00 | LMS (↑) | ICAT (↑) |
> |-------------------------------|-----------|---------|----------|
> | BERT-Base-cased               | 56.93     | 87.29   | 75.19    |
> | BERT-Base-cased + Intersection_IG | 63.88 | 75.39   | 54.46    |
>
> **Q6: Table 5, it should be "StereoSet".**
>
> **A6**:
> Thank you for pointing this out! We have corrected it in the revised paper.

---

### Meta-Review · Area_Chair_fWxU · 2023-12-08

**Metareview:**

This paper addresses the critical issue of bias in language models (LMs) and proposes an innovative approach, integrated gap gradients (IG^2), to identify and mitigate social bias in neurons within LMs. The positive reviews highlight the paper's clarity, ease of implementation, and its contribution to the important area of social bias in neural networks. The reviewers commend the paper for combining interpretability techniques with practical applications in social bias research, noting the potential for the proposed method to lay a foundation for future work in natural language processing. The experiments are deemed thorough, and the overall writing is considered well-structured and accessible.

On the other hand, questions were raised about the generalizability of IG^2, particularly beyond predicting sensitive attributes, and the potential limitations of focusing solely on masked language modeling tasks for experiments. Additionally, reviewers suggest exploring the implications of the results in more depth and consider alternative approaches, such as regularizing detected neurons rather than suppressing them entirely.

Despite these considerations, the overall positive impact, clarity, and significance of the paper are recognized by all reviewers. The suggested improvements, including addressing generalizability concerns,  and exploring broader applications, would enhance the paper.

**Justification For Why Not Higher Score:**

Most of the reviewers stayed at borderline accept after the discussion phase. Hence AC felt that this paper shall be accepted as a poster.

**Justification For Why Not Lower Score:**

All five reviewers offered positive scores for this paper.

---

### Decision · Program_Chairs · 2024-01-16

Accept (poster)